# Effect of Air Velocity and Initial Conditioning on the Moisture Buffer Value of Four Different Building Materials

**DOI:** 10.3390/ma16083284

**Published:** 2023-04-21

**Authors:** Sana Khaled, Florence Collet, Sylvie Prétot, Marjorie Bart

**Affiliations:** Laboratoire de Génie Civil et Génie Mécanique LGCGM, Univ. Rennes, 35704 Rennes, France; sana.khaled@univ-rennes.fr (S.K.); sylvie.pretot@univ-rennes.fr (S.P.); marjorie.bart@univ-rennes.fr (M.B.)

**Keywords:** moisture transfer, MBV test, NORDTEST protocol, air velocity, initial conditioning, gypsum, cellular concrete, thermo-hemp, fine-hemp

## Abstract

Porous materials are able to exchange moisture with the surrounding air. The more hygroscopic they are, the more they contribute to regulate ambient humidity. This ability is characterized by the moisture buffer value (MBV) which is measured under dynamic solicitations according to different protocols. The NORDTEST protocol is the most commonly-used. It gives recommendations regarding the air velocity and the ambient conditions for initial stabilization. The purpose of this article is to measure the MBV according to the NORDTEST protocol and to study the effect of air velocity and of initial conditioning on the MBV results for different materials. Two mineral and two bio-based materials are considered: gypsum (GY), cellular concrete (CC), thermo-hemp (TH) and fine-hemp (FH). Following the NORDTEST classification, GY is a moderate hygric regulator, CC is good, TH and FH are excellent. When the air velocity ranges from 0.1 to 2.6 m/s, the MBV of GY and CC materials remains constant, but the MBV of TH and FH materials is highly affected. The initial conditioning has no effect on the MBV, but has an effect on the water content of the material, whatever the material.

## 1. Introduction

Heat and moisture transfers in porous building materials widely impact the building’s energy efficiency, indoor climate and occupants’ health. These transfers depend on the hygrothermal properties of building materials. Among them, the sorption isotherm and the vapor permeability are measured under steady-state conditions. The moisture buffer value (MBV) is obtained from dynamic solicitations. It characterizes the capacity of a material to moderate the hygric variations of the ambient air. 

Krieger et al. [1] provide a state-of-the-art study on MBV characterization methods. According to them, the most commonly used method is the NORDTEST protocol [2], which is employed in more than 70% of the studies. In this method, the specimens are exposed to dynamic cycles of 8 h at 75%RH followed by 16 h at 33%RH at 23 °C. The specimen thickness should be greater than the penetration depth. The specimens should be stabilized at 23 °C and 50 ± 5%RH prior to the MBV test. The air velocity should be equal to 0.1 ± 0.05 m/s near the exchange surfaces. The NORDTEST protocol classifies the ability of materials to regulate the ambient relative humidity according to their MBV. Strongly hygroscopic materials, such as bio-based materials, with high storage capacity and/or high vapor permeability have MBV above 1 g/(m^2^.%RH) and are classified as good to excellent hygric regulators [3,4,5,6,7,8,9,10,11]. Materials with low storage capacity and low vapor permeability such as gypsum lead to low MBV and are classified as moderate humidity regulators [12,13,14]. Table 1 summarizes the MBV results and test conditions of investigations on materials that can be compared to those studied in this paper: gypsum, cellular concrete and bio-based materials, e.g., hemp concrete.

Tran Le [15], Evrard [16], and Rahim et al. [3] studied the effect of specimen thickness on MBV of hemp concrete material. They all found that the MBV is underestimated when the thickness is not large enough, as shown in Table 1. For example, according to the literature, the MBV of hemp concrete ranges from 1.90 to 2.40 g/(m^2^.%RH) [4,5,17,18]. Evrard [16] measured the MBV of hemp concrete on 1.75 cm thick specimens. The MBV found was 1.39 g/(m^2^.%RH), which is lower than the expected value.

Regarding the air velocity, some authors did not mention its value during MBV testing [12,15,19,20,21,22] (Table 1). The MBV measured on hemp concrete in [19,20] (2.78–3.41 g/(m^2^.%RH)) were higher than the values measured at 0.1 m/s. Latif et al. [23] obtained a high MBV of 3.47 g/(m^2^.%RH) with a high discrepancy for hemp lime which they explained by the inhomogeneity and the variation of air velocity. Cascione et al. [24] found that the increase in air velocity from 0.23 to 0.77 m/s resulted in an increase in the MBV of clay plaster from 1.49 to 1.82 g/(m^2^.%RH). For gypsum board samples, painted or not, Shi et al. [25] found that when the air flow rate in the test chamber increased, the MBV increased. The study of Holcroft et al. [26] was based on ISO 24353. They found that the MBV increased from 1.9 to 4.8 g/(m^2^.%RH) as the air velocity increased from 0.1 m/s to 0.6 m/s.

Lastly, regarding the influence of initial conditions, some authors did not mention or consider the initial stabilization (23 °C; 50 %RH) within their studies as shown in Table 1. Osanyintola et al. [27] concluded that the initial conditions do not affect the MBV results for a spruce plywood material but they affect the mass change during the first few cycles. Abdellatef et al. [20] found that the initial moisture condition did not have a significant effect on the MBV of different hempcrete compositions, with a difference of less than 1%.

To sum up, thickness affects MBV results in link with penetration depth. Air velocity influences the MBV results while initial moisture conditions seem to have no effect. This study focuses on the hygric behavior at the material scale, so it is performed with a thickness higher than the penetration depth. The effect of air velocity and initial conditioning is investigated. For a reliable comparison between four construction materials, these are tested simultaneously, both for stabilization and for cyclic loads.

Biotic and abiotic materials are chosen: gypsum (GY), cellular concrete (CC), thermo-hemp (TH) and fine-hemp (FH). Gypsum and cellular concrete are both known for their low hygroscopicity. These materials have low storage capacity as indicated in [12,14] for gypsum and in [28,29,30,31,32] for cellular concrete. They have low vapor permeability between 1.4 × 10^−11^ and 2.4 × 10^−11^ kg/(m.s.Pa) for gypsum material [13] and about 2 × 10^−11^ to 3 × 10^−11^ kg/(m.s.Pa) for cellular concrete [30,31,32,33]. Hemp concretes are highly hygroscopic, with high storage capacity and high water vapor permeability [5,17,34]. Firstly, our methods and materials are provided. Then, after presenting the results obtained following the NORDTEST protocol, the effects of air velocity and initial conditioning are detailed. 

**Table 1 materials-16-03284-t001:** Summary of the different parameters, dynamic solicitations and MBVs for different materials presented in the literature.

Materials	Apparent Density [kg/m^3^]	Thickness of Specimens [cm]	Air Velocity [m/s]	Initial Conditions	MBV [g/(m^2^.%RH)]	References
Gypsum	1000	1.25–1.30	0.1 ± 0.05	23 °C; 50%RH	0.63	[2]
Gypsum	1200	2.00	-	-	0.51–0.72	[12]
Gypsum plaster	-	10.00 3.00 1.00 0.50	-	-	0.94 0.91 0.43 0.22	[21]
Cellular concrete	500	0.74–0.75	0.1 ± 0.05	23 °C; 50%RH	1.04	[2]
Cellular concrete	540	7.00–8.00	Vertical velocity 0.07–0.14 and horizontal velocity: 0.1–0.4	23 °C; 50%RH	1.14–1.42	[4]
Aerated cellular concrete	-	10.00 3.00 1.00 0.50	-	-	0.81 0.82 0.61 0.33	[21]
Hemp concrete	412 451	7.40–8.30	Vertical velocity 0.07–0.14 and horizontal velocity: 0.1–0.4	23 °C; 50%RH	2.08 2.22	[4]
Hemp concrete	450	11.50	-	23 °C; 50%RH	1.90	[16]
Hemp concrete	-	5.00	-	23 °C; 50%RH	3.05	[19]
Hempcrete	290–310	5.50	-	-	2.78	[20]
Sprayed hemp concrete	450 ± 20	10.00	Not exceed 0.30	Dried at 60 °C than stabilized at 23 °C; 50%RH	2.30	[35]
Precast hemp concrete	460	7.00–8.00	Vertical velocity 0.07–0.14 and horizontal velocity: 0.1–0.4	23 °C; 50%RH	1.94	[5]
Sprayed and molded hemp concrete	430	7.00–8.00	Vertical velocity 0.07–0.14 and horizontal velocity: 0.1–0.4	23 °C; 50%RH	2.14–2.15	[5]
Moulded hemp concrete	320	3.00 6.00	-	-	1.79 1.99	[15]
Lime hemp concrete	400	7.50	0.10	55%RH	2.36	[17]
Hemp lime concrete	478 ± 7	3.00 5.00 7.00	-	Dried at 60 °C	1.84 1.86 2.02	[3]
Hemp lime	290	12.00	0.15–0.20	23 °C; 50%RH	3.47	[23]
Hemp starch composites	170 210	2.50	-	23 °C; 50%RH	2.52 3.49	[11]
Hemp straw composites	166 188	-	0.1 to 0.4 for horizontal velocity and lower than 0.15 for vertical one	23 °C; 50%RH	2.20 2.42	[10]
Hemp composites with mineral or organic binder	$\rho_{\mathrm{PLA}\;\mathrm{HC}\;}$ = 250–500 $\rho_{\mathrm{prompt}\;\mathrm{cement}\;\mathrm{HC}}$= 500 $\rho_{\mathrm{lime}\;\mathrm{based}\;\mathrm{HC}\;}$ = 460	-	-	Dried at 60 °C, then stabilized at 23 °C; 50%RH	MBV _PLA HC_ = 1.77 MBV _prompt cement HC _= 2.17 MBV _lime based HC_ = 1.92	[22]
Hemp stabilized clay	373 510	7.00	Vertical velocity 0.07–0.14 and horizontal velocity: 0.1–0.4	23 °C; 50%RH	2.24 2.33	[18]
Clay plaster	1258	2.00	0.23–0.77	23 °C; 60%RH	1.49–1.82	[36]

## 2. Methods and Materials

### 2.1. MBV Experimental Method and Device

#### 2.1.1. Recommendations of NORDTEST Protocol

The NORDTEST method requires at least three specimens per material with a minimum exchange area of 100 cm^2^ each and a total exchange surface of 300 cm^2^. All the other sides of the specimens are sealed. The thickness of the specimen should be greater than the penetration depth. The air velocity should be 0.10 ± 0.05 m/s near the exchange surfaces. Prior to the test, the specimens were stabilized at 23 ± 5 °C; 50 ± 5% RH. After stabilization, the specimens were exposed to a humidity variation of 8 h at 75% RH followed by 16 h at 33% with a constant temperature of 23 °C. Cycles were repeated until the change in mass was less than 5% between the last three cycles. Specimens should be weighed at least once during each cycle and at least 5 times during the 8 h at high RH over the last three cycles. 

For each cycle, during adsorption or desorption, the MBV [g/(m^2^.%RH)] was calculated as follows [2]:(1)$$\mathrm{MBV}=\frac{\Delta m}{A\left({\mathrm{RH}}_{\mathrm{high}}-{\mathrm{RH}}_{\mathrm{low}}\right)}$$
with Δm [g] the moisture uptake/release during cycles, A [m^2^] the exposed surface area, RH_high/low_ [%] the high and low relative humidity levels.

The average MBV between adsorption and desorption was calculated for the last three cycles. Then the materials were classified following the NORDTEST classification [2]:MBV < 0.2 g/(m^2^.%RH): negligible0.2 < MBV < 0.5 g/(m^2^.%RH): limited0.5 < MBV < 1.0 g/(m^2^.%RH): moderate< MBV < 2.0 g/(m^2^.%RH): good2 g/(m^2^.%RH) < MBV: excellent

#### 2.1.2. Study following the NORDTEST Protocol

In this study, three cylindrical specimens per material were used. The total exchange surface of the three specimens was about 300 cm^2^. Their thickness was greater than the penetration depth found in the literature: 3.3 cm for gypsum [2], 5.2 cm for cellular concrete [2], 5.8 cm for thermo-hemp [5,16], and 3.0 cm for fine-hemp [37]. All the specimens were sealed with aluminum tape on the bottom and lateral sides to ensure a unidirectional vapor transfer.

Prior to the test, all the specimens were dried and then stabilized at 23 °C; 50%RH. The drying was performed at a suitable temperature to prevent the specimens from damage. GY specimens were dried at 40 °C according to NF EN 12860 [38] and others at 60 °C according to RILEM TC 236-BBM recommendations [39]. The stabilization of specimens at different humidities was performed in a climatic chamber VÖtsch^®^ VC4060, with a temperature range from 10 to 95 °C and a relative humidity range from 10 and 98%. The drying and stabilization criteria were calculated as follows:(2)$${\left(\frac{\Delta m}{m}\right)}_{24h}=\frac{\frac{m_{i+1}-m_{i}}{m_{i}}}{t_{i+1}-t_{i}}\;(\%/24\;h)$$
where m [g] is the weight of the specimen, t [h] is the weight time, and i and i + 1 are the day and the following day, respectively.

The dry point is reached when the criterion is lower than 0.1%/24 h according to NF EN ISO 12571 [40]. The wet point is stabilized when the criterion is lower than 0.01%/24 h.

After stabilization, the specimens were exposed to daily cyclic variations: 8 h at high relative humidity (75%) followed by 16 h at low relative humidity (33%) at 23 °C. During the test, they were regularly weighed outside the climatic chamber: five times during the adsorption phase and two times during the desorption phase.

The ambient relative humidity and temperature in the climatic chamber were measured with Sensirion SHT75 sensor every 180 s. The weighing was performed with Explorer EX1103 weighing scales with an accuracy of 0.004 g. The air velocity was measured at 1 cm above the center of the specimen exchange surface using an omnidirectional anemometer connected to a Multilogger M1300 acquisition box. The value of air velocity was measured each 1 s and averaged over 1 min. Some grids were used to limit and homogenize the airflow inside the climatic chamber (Figure 1). 

#### 2.1.3. Modified Protocols

In order to study the effect of air velocity, three tests were performed. Test 1 targeted the air velocity of the NORDTEST protocol recommendations. Tests 2 and 3 were performed by using fans to reach higher air velocities (Figure 2). The three tests were run consecutively with two intermediate cycles where the weighing was performed only when the set point changes. In order to study the effect of initial conditioning, three tests were conducted by stabilizing the specimens at 33%RH (test 4), 50%RH (test 5) and finally at 65%RH (test 6).

Table 2 summarizes the steps followed when studying the effect of the two parameters on MBV results.

Despite the differences with the NORDTEST protocol, the MBV expression and the NORDTEST classifications of Rode et al. [2] were used.

### 2.2. Materials 

Four materials were studied: two mineral, homogeneous and isotropic materials (gypsum and cellular concrete) and two anisotropic and bio-based materials (thermo-hemp and fine-hemp). 

#### 2.2.1. Production of Specimens

Gypsum (GY) and cellular concrete (CC) specimens were obtained by coring commercial blocks as shown in Figure 3. Thermo-hemp (TH) and fine-hemp (FH) were manufactured according to the protocol described below. Two sizes of specimens were considered (Table 3). For all the experimental measurements, the size of the specimen was greater than the representative elementary volume (REV). The criterion for REV was 2.5 times the characteristic length, denoted as l [39]. For GY and CC, their REV was defined from the size of particles and pores. The GY material had a particle size < 0.2 mm [41,42]. The CC material had large artificial pores with a radius between 0.01 and 1 mm [28,29,30,43]. The hemp concrete had a representative elementary volume (REV) about 100 cm^3^ (side > 4.7 cm) according to Collet et al. [44] and Evrard [16]. This value is considered for TH and FH materials. For the MBV test, the exchange surface area of each specimen was higher than 100 cm^2^ and its height was greater than its penetration depth, as given in Section 2.1.2.

The hemp materials were made of a binder (Thermo^®^ or Stabilized Fines), hemp and water. The formulations are given in Table 4. For TH, the binder was Thermo^®^ and the hemp shiv was pre-wet with water to hemp mass ratio equal to 0.40. For FH, the binder consisted of stabilized fines made from washing mud and stabilized with 10% Thermo^®^ [18]. To reduce the energy consumption, the muds were not dried, a target density was used to reach the same water content of the paste [45].

For the production (Figure 4), binder paste was prepared by mixing its components for about 2 min. Then, hemp shiv was added gradually into the binder paste and the mixing was carried on for about 5 min. After mixing, the specimens were produced in two layers of about 5 cm each, by compacting them at 0.1 MPa with an automatic press.

The specimens were then placed in a laboratory room; a lid was placed on top of each mold to dry the specimen without drying its surface too quickly. After three days, the specimens were demolded and stabilized at ambient conditions.

#### 2.2.2. Density and Porosity

The dimensions of specimens were measured using a caliper with an accuracy of 0.01 mm. The apparent density $\rho_{app}$ [kg/m^3^], was calculated from the weight after the initial stabilization at 23 °C; 50%RH, and at dry state. The real density,$\;\rho_{r}$ [kg/m^3^], was measured by the pycnometer method with toluene according to NF EN ISO 11508 [46]. The samples were ground and sieved with a 0.5 mm mesh sieve using a Retsch ZM 200 mill. Then, they were poured into five pycnometers with a volume of 100 mL each. The total porosity [%] was calculated from the real density and the dry apparent density of the material. The open porosity was calculated from hydrostatic weighing, except for FH for which it is not possible.

Table 5 shows the mean values and the standard deviations of densities [kg/m^3^] and total and open porosities [%] of the materials. The apparent density of GY was around 1000 kg/m^3^. For CC, TH and FH, the apparent densities were lower, between 350 and 515 kg/m^3^. Both mineral materials, GY and CC, had values of real density about 2600 kg/m^3^. Bio-based materials had close real densities, around 1800 kg/m^3^. For all materials, the porosity was mainly open. For GY, the total and open porosities were about 60% and 55%. CC, TH and FH had higher total porosities, around 80%. The open porosity of CC was lower than for TH, at 72% and 76%, respectively.

These results are in agreement with values from literature. For gypsum materials, the real density was between 2300 and 2600 kg/m^3^, depending on its hydration [47,48] the apparent density ranges between 940 and 1300 kg/m^3^ [49,50,51,52], and the open porosity ranges from 36% to 68%, depending on water/gypsum mass ratio and on hydration [52,53,54]. For cellular concretes, the apparent density ranges from 300 to 600 kg/m^3^, the real density ranges around 2500 and 2600 kg/m^3^, the total and open porosities are close and are about 69%–87% [28,29,30,55,56]. For bio-based materials with a hemp to binder mass ratio of 0.5, the apparent density ranges from 390 to 450 kg/m^3^, the real density is between 1800 and 2000 kg/m^3^, the total and open porosities range from 72% to 85% and 54% to 79% [5,18,57,58,59].

## 3. Results

### 3.1. Experimental Conditions

#### 3.1.1. Stabilization Time

Depending on the hygric history of the specimens, the time required to reach the stabilization criterion varies. Drying takes 2 to 3 days for GY and CC, and 3 to 7 days for TH and FH. Regardless of the RH conditions, the stabilization time for GY is always about 2 to 3 days. As shown in Table 6, for CC, the stabilization times range from 2 to 18 days. For TH and FH, the stabilization times are longer and range from 3 to 10 weeks. In this study, all the specimens were studied simultaneously so they are all conditioned at the same time, corresponding to the longer one.

#### 3.1.2. Ambient Conditions Control

The ambient experimental conditions are well controlled in the climate chamber, as shown in Figure 5, displaying the RH [%] and T [°C] recorded during test 6. All cycles are repeatable and very similar, for RH ads. and RH des., the variance is less than 3% (Table 7). The peaks displayed in Figure 5 correspond to the drop in relative humidity and temperature caused by the regular opening of the door for the weighing process. During each test, it takes about 15–30 min to increase from 33% to 75%, and about 15–50 min to decrease from 75% to 33%. Although NORDTEST recommends that the set point change should be performed within 30 min, Roels and Jansen [21] have shown that 1.5 h or less has no effect on the MBV results.

#### 3.1.3. Air Velocity Control

The air velocity conditions are well controlled. For test 1, the average air velocity is of 0.102 m/s with a standard deviation of ±0.011 m/s for all specimens. It meets the NORDTEST recommendations. 

For the modified protocols, the air velocity ranges between 0.376 m/s and 2.589 m/s with a relative deviation of 1.0% to 7.0% for the second test, and 2.5% to 8.0% for the third test. The standard deviation increases with air velocities.

Air velocities values are detailed in Section 3.3 with MBV results.

### 3.2. NORDTEST Project

For each material, the kinetic of mass change of one specimen is given in Figure 6 as the three specimens follow similar pattern of moisture adsorption and desorption. Along the five cycles, the mass beams show a decreasing tendency for CC, TH and FH and a slightly increasing one for GY. The steady state is reached from the third cycle. The average mass change calculated from cycles 3 to 5 is equal to 0.20 g for GY material, 0.40 g for CC material, 0.80 g for TH material and 0.90 g for FH materials.

The MBV obtained from the three specimens of each material are 0.53 g/(m^2^.%RH) for GY, 1.23 g/(m^2^.%RH) for CC, 2.12 g/(m^2^.%RH) for TH and 2.38 g/(m^2^.%RH) for FH. The higher the open porosity is the higher the MBV is. According to the NORDTEST classification, GY and CC are classified as moderate and good humidity regulators while TH and FH are classified as excellent humidity regulators (Figure 7). Bio-based materials show a higher capacity to regulate ambient relative humidity compared to GY and CC. These results are in good agreement with the results from the literature (Table 1).

### 3.3. Influence of Air Velocity

The moisture uptake and release of one specimen per material are displayed in Figure 8, Figure 9, Figure 10 and Figure 11 during the three tests (1, 2 and 3). Whatever the air velocity value, the steady state is reached from the third cycle. Table 8 gives the MBVs under adsorption, desorption and in average for each specimen per material. Figure 12 displays the average MBV for each specimen per material according to the air velocity. 

Both for GY and CC, the kinetics are similar for the three tests, regardless of air velocity (Figure 8 and Figure 9). The mass change is constant and ranges from 0.21 to 0.23 g for GY and from 0.38 to 0.41 g for CC. The MBV of GY and CC remains constant, at 0.53 ± 0.02 g/(m^2^.%RH) and 1.25 ± 0.04 g/(m^2^.%RH), respectively, as shown in Figure 12. 

For TH and FH, the amplitude of mass beam increases significantly with air velocity. For example, for TH-c-7, raising air velocity from 0.1 m/s to 1.0 m/s results in an increase in mass change from 0.78 g to 1.69 g, and an increase in MBV of about two-fold. For FH-c-4, raising the air velocity from 0.1 m/s to 2 m/s results in an increase in mass change from 0.92 g to 2.40 g, and MBV is multiplied by about two and a half times. The variation of MBV versus air velocity for TH and FH shows scattered upward trends. Even if the variation is significant, it does not allow us to identify a variation law.

The MBV is related to the hygroscopicity of the material and to the surface transfer coefficient that depends on the air velocity and on the roughness of the exchange surface. The mass transfer coefficient increases with the heat transfer one as defined in Lewis relation [60] and the heat transfer coefficient increases with air velocity depending on the surface roughness as presented in McAdams model [61]. In a numerical study, Asli et al. [62] found that when the surface transfer coefficients increases from 1.10^−8^ s.m^−1^ to 3.10^−6^ s.m^−1^, the MBV of hemp concrete increases from 1.82 to 3.02 g/(m^2^.%RHAs GY and CC have low hygroscopicity and smooth exchange surface, their MBV remains constant. Given that TH and FH have a high storage capacity and high water vapor permeability as well as having a rough exchange surface with large voids means their MBV increases significantly.

### 3.4. Influence of Initial Conditioning

The kinetics of mass of one specimen per material is given in Figure 13, Figure 14, Figure 15 and Figure 16. For tests 1 and 4 to 6, Table 9 displays the initial and final water contents, the average mass changes over cycles 3 to 5, and the MBV for the four materials.

The initial water content impacts the trend of mass beam. For specimens having low initial water content (test 4—stabilized at 33%RH), the kinetics of all materials show an increasing tendency. The materials absorb more moisture than they release up to reach steady cycles. This induces the increase in water content during MBV test (Table 9). For specimens having high initial water content (test 6—stabilized at 65%RH), the reverse occurs. For tests 1 and 5, the kinetics of mass are almost overlapped for the four materials. They differ slightly due to the light difference in initial water content induced by the hygric background. Moreover, the kinetics of mass are almost horizontal as the final water content is close to the initial one.

The average mass change in the last three cycles is similar regardless of the initial water content for the four materials. The mass change is around 0.20 g for GY, about 0.37 g for CC, 0.73–0.78 g for TH and 0.82–0.88 g for FH. Hence, as found in the literature the MBV remains unchanged whatever the initial condition [20,27], however it affects the water content during the test [27].

## 4. Conclusions

This study investigates the impact of air velocity and initial conditioning on MBV. The results of two mineral and homogeneous materials with smooth exchange surfaces (gypsum and cellular concrete) and on two bio-based materials with rough exchange surfaces (thermo-hemp and fine-hemp) are compared.

The measurements carried out following the NORDTEST recommendations classify GY as moderate hygric regulator, CC as good, TH and FH as excellent.

Air velocity has no effect on MBV for GY and CC, but a significant effect for TH and FH. For bio-based materials, the MBV increase is assigned to their hygroscopicity and to the mass transfer coefficient that may increase with the air velocity and the surface roughness. Hence, to compare MBV results between materials, tests must be performed at the same air velocity. The value of 0.1 m/s given in the NORDTEST protocol leads to hygric buffering representative for indoor ambient moisture conditions.

For all studied materials, initial conditioning has no effect on MBV but it affects the water content throughout the MBV test. Even if the MBV relates the ability of materials to absorb and desorb moisture, it is not sufficient to fully characterize the hygric behavior of materials.

## Figures and Tables

**Figure 1 materials-16-03284-f001:**
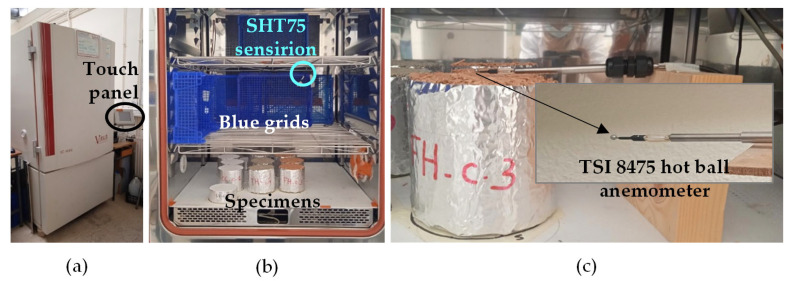
MBV experimental setup: (**a**) climatic chamber; (**b**) air velocity control (**c**) hot ball anemometer.

**Figure 2 materials-16-03284-f002:**
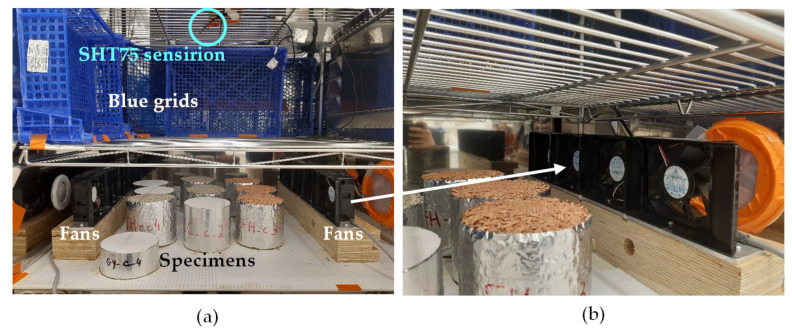
MBV experimental setup: (**a**) air velocity control; (**b**) fans.

**Figure 3 materials-16-03284-f003:**
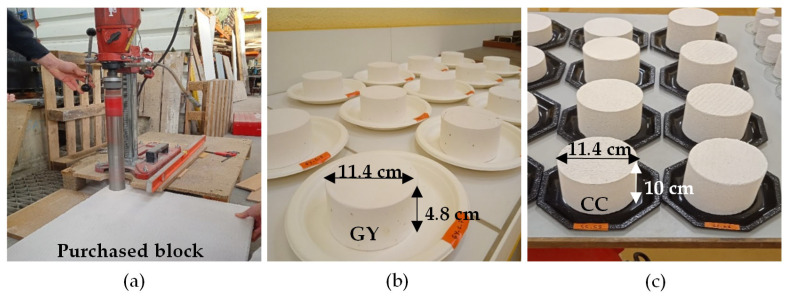
(**a**) Coring the purchased blocks; (**b**) gypsum specimens; (**c**) cellular concrete specimens.

**Figure 4 materials-16-03284-f004:**
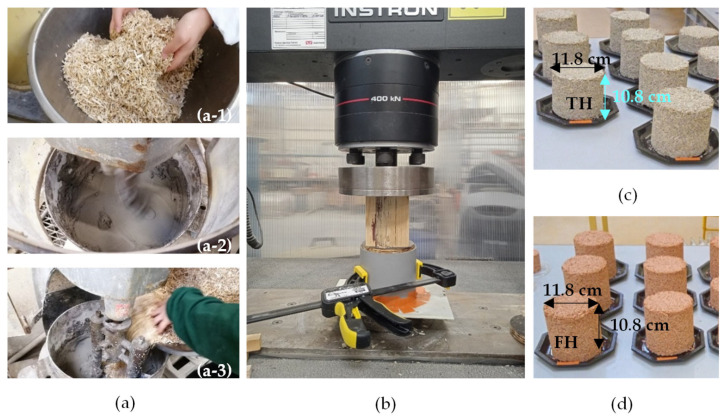
Mixing procedure of thermo-hemp and fine-hemp: (**a**-**1**) manual mixing of hemp shives with water; (**a**-**2**) mixing of binder with water; (**a**-**3**) mixing all components; (**b**) compaction; (**c**) thermo-hemp; (**d**) fine-hemp specimens.

**Figure 5 materials-16-03284-f005:**
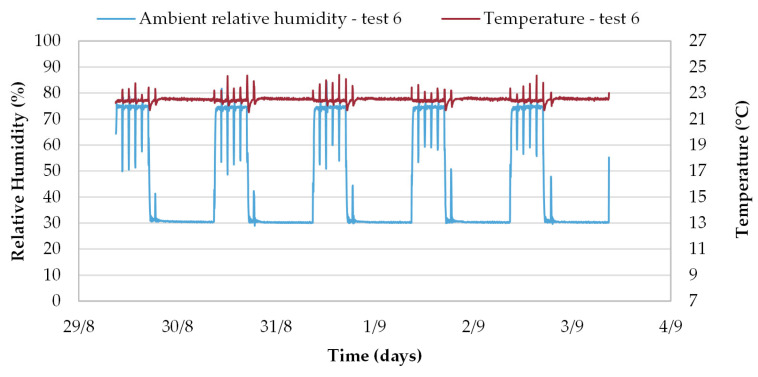
Ambient relative humidity [%], and temperature [°C] recorded during test 6 of MBV.

**Figure 6 materials-16-03284-f006:**
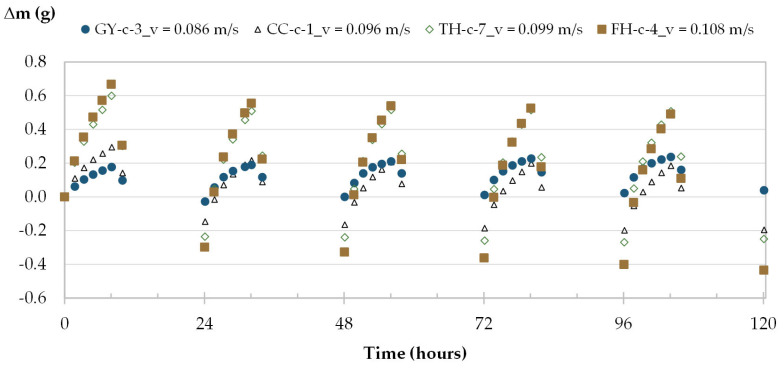
Moisture uptake and release for one specimen of each material along test 1 of MBV.

**Figure 7 materials-16-03284-f007:**
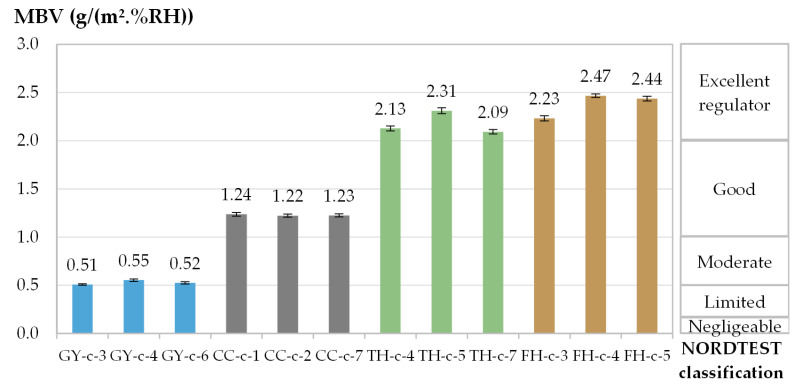
Average MBV [g/(m^2^.%RH)] and standard deviation obtained for each specimen per material along test 1 and NORDTEST classification [2].

**Figure 8 materials-16-03284-f008:**
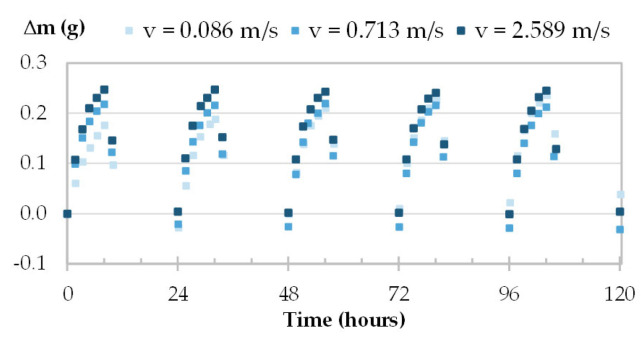
Moisture uptake and release kinetics for one specimen of GY (GY-c-3) in tests 1-2-3.

**Figure 9 materials-16-03284-f009:**
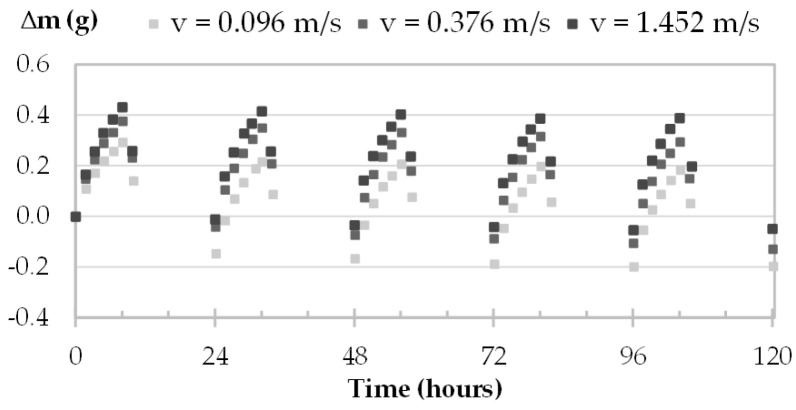
Moisture uptake and release kinetics for one specimen of CC (CC-c-1) in tests 1-2-3.

**Figure 10 materials-16-03284-f010:**
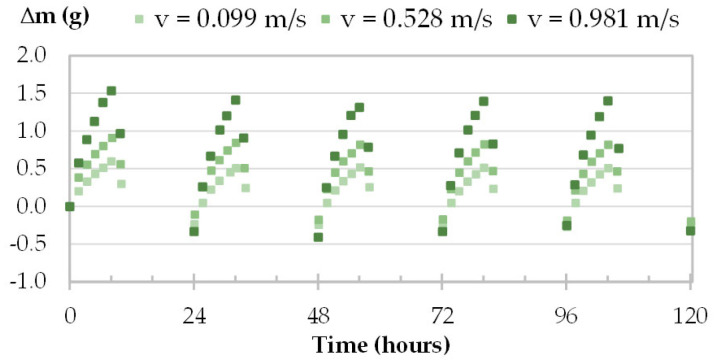
Moisture uptake and release kinetics for one specimen of TH (TH-c-7) in tests 1-2-3.

**Figure 11 materials-16-03284-f011:**
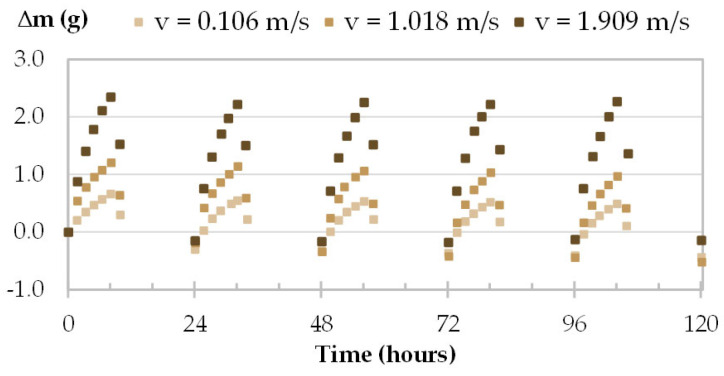
Moisture uptake and release kinetics for one specimen of FH (FH-c-4) in tests 1, 2 and 3.

**Figure 12 materials-16-03284-f012:**
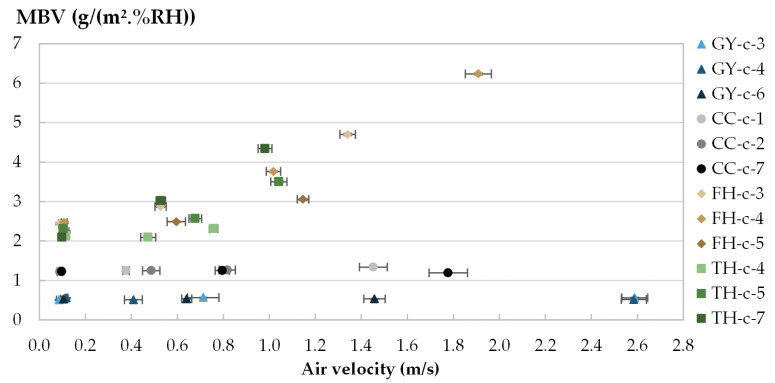
MBV versus air velocities for each specimen.

**Figure 13 materials-16-03284-f013:**
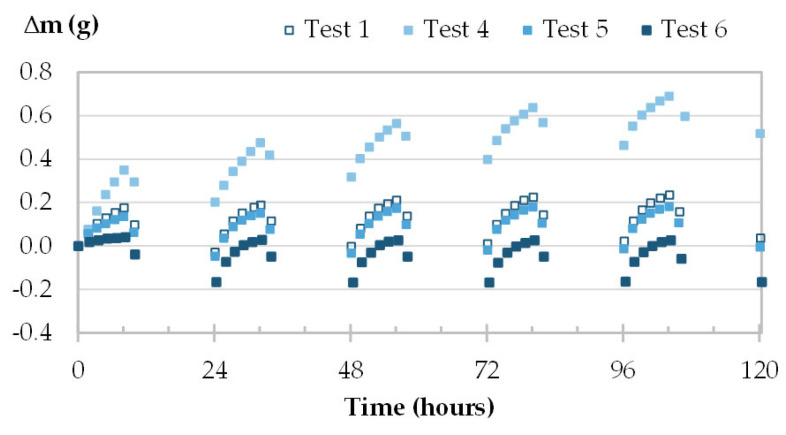
Moisture uptake and release for one specimen of GY (GY-c-3) at different initial conditioning.

**Figure 14 materials-16-03284-f014:**
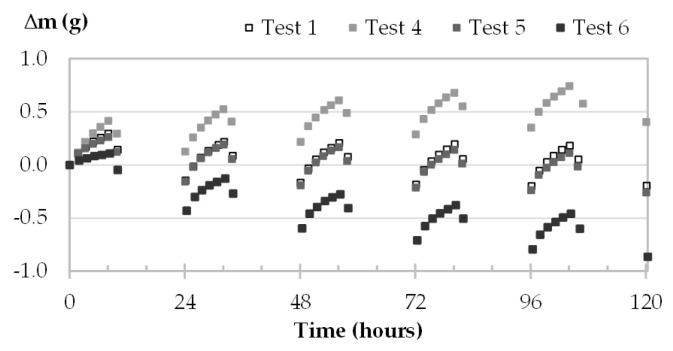
Moisture uptake and release for one specimen of CC (CC-c-1) at different initial conditioning.

**Figure 15 materials-16-03284-f015:**
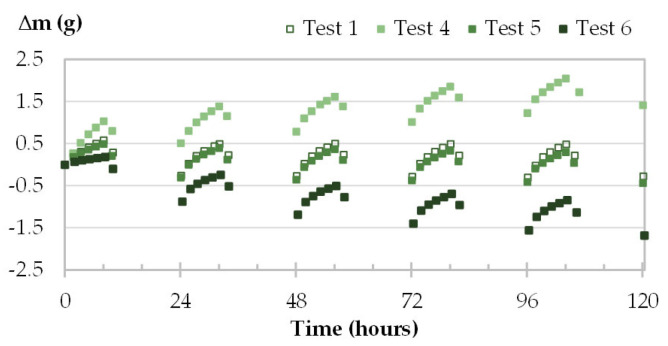
Moisture uptake and release for one specimen of TH (TH-c-4) at different initial conditioning.

**Figure 16 materials-16-03284-f016:**
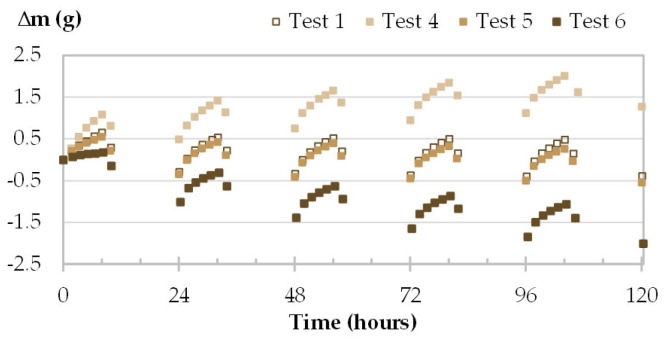
Moisture uptake and release for one specimen of FH (FH-c-3) at different initial conditioning.

**Table 2 materials-16-03284-t002:** Progress of the experimental study.

	Effect of Air Velocity					Effect of Initial Conditioning		
	Test 1 (NORDTEST)	Intermediate Cycles	Test 2	Intermediate Cycles	Test 3	Test 4	Test 5	Test 6
Number of cycles	5	2	5	2	5	5		
Air velocities	0.1 m/s	Follow the first test conditions	Higher air velocities	Follow the second test conditions	Higher air velocities	0.1 m/s		
Initial conditions	Dried then stabilized at 23 °C; 50%RH		Follow test 1		Follow test 2	Dried then stabilized at 23 °C; 33%RH	23 °C 50%RH	23 °C 65%RH

**Table 3 materials-16-03284-t003:** Size of specimens for density, porosity and MBV tests.

	For MBV Test			For Density and Porosity Measurement
	GY	CC	TH and FH	GY, CC, TH and FH
h [mm]	48	100	108	76
d [mm]	114	114	118	57

**Table 4 materials-16-03284-t004:** Formulation of thermo-hemp (TH) and fine-hemp (FH) composites in mass portions with reference to the mass of hemp.

	Thermo-Hemp (TH)	Fine-Hemp (FH)
T: Thermo	2.00	0.18
F: Fines	-	1.82
H: Hemp Biofibat	1.00	1.00
W_T_: Total water	1.60	1.60

**Table 5 materials-16-03284-t005:** Mean values and standard deviations of densities [kg/m^3^] and total and open porosities [%] of the studied materials.

Materials	Apparent Density_23°C;50%RH_ [kg/m^3^]	Dry Apparent Density [kg/m^3^]	Real Density [kg/m^3^]	Total Porosity [%]	Open Porosity [%]
GY	1005 ± 10	1002 ± 8	2611 ± 20	62.17	55.55 ± 0.67
CC	518 ± 5	513 ± 5	2602 ± 53	81.34	72.38 ± 1.58
TH	425 ± 9	408 ± 15	1752 ± 54	76.53	76.22 ± 1.77
FH	364 ± 9	350 ± 8	1866 ± 49	81.30	-

**Table 6 materials-16-03284-t006:** Stabilization times for tests 1, 4, 5 and 6 for CC, TH and FH.

	Test 1	Test 4	Test 5	Test 6
CC	17 days	18 days	2 days	2 days
TH	10 weeks	7 weeks	4 weeks	9 weeks
FH	9 weeks	5 weeks	3 weeks	7 weeks

**Table 7 materials-16-03284-t007:** Average relative humidity [%] and temperature [°C] for all cycles (uptake/release) during each MBV test.

	Effect of Air Velocity			Efect of Initial Water Content		
	Test 1	Test 2	Test 3	Test 4	Test 5	Test 6
**HR_ads._ [%]**	69.01	68.67	67.77	69.74	69.93	72.67
**HR_des._ [%]**	29.18	27.03	26.15	29.02	28.43	30.95
**T_ads.&des._ [°C]**	21.83	22.29	22.23	22.50	22.68	22.48

**Table 8 materials-16-03284-t008:** Density and MBV (for adsorption and desorption phases and in average) at different air velocities for each specimen.

Specimens and Density_(23°C;50%RH)_ [kg/m^3^]	Air Velocities [m/s]	MBV Ads. [g/(m^2^.%RH)]	MBV Des. [g/(m^2^.%RH)]	MBV Av. [g/(m^2^.%RH)]
GY-c-3 1000.2	0.086 ± 0.011	0.52 ± 0.01	0.49 ± 0.01	0.51 ± 0.01
	0.713 ± 0.068	0.56 ± 0.01	0.57 ± 0.01	0.56 ± 0.01
	2.589 ± 0.056	0.56 ± 0.01	0.56 ± 0.01	0.56 ± 0.01
GY-c-4 997.5	0.118 ± 0.014	0.58 ± 0.02	0.53 ± 0.01	0.55 ± 0.01
	0.409 ± 0.022	0.51 ± 0.01	0.51 ± 0.00	0.51 ± 0.00
	2.585 ± 0.046	0.52 ± 0.01	0.52 ± 0.02	0.52 ± 0.01
GY-c-6 1015.4	0.106 ± 0.006	0.54 ± 0.01	0.51 ± 0.01	0.52 ± 0.01
	0.640 ± 0.039	0.53 ± 0.01	0.53 ± 0.00	0.53 ± 0.00
	1.457 ± 0.054	0.54 ± 0.01	0.53 ± 0.01	0.53 ± 0.00
CC-c-1 521.5	0.096 ± 0.006	1.22 ± 0.02	1.25 ± 0.03	1.24 ± 0.02
	0.376 ± 0.014	1.22 ± 0.01	1.28 ± 0.01	1.25 ± 0.01
	1.452 ± 0.061	1.33 ± 0.03	1.35 ± 0.01	1.34 ± 0.01
CC-c-2 513.0	0.089 ± 0.012	1.21 ± 0.02	1.24 ± 0.03	1.22 ± 0.02
	0.487 ± 0.038	1.22 ± 0.00	1.28 ± 0.01	1.25 ± 0.00
	0.818 ± 0.035	1.26 ± 0.02	1.27 ± 0.02	1.26 ± 0.01
CC-c-7 520.1	0.097 ± 0.008	1.21 ± 0.01	1.24 ± 0.03	1.23 ± 0.02
	0.796 ± 0.031	1.22 ± 0.01	1.28 ± 0.07	1.25 ± 0.00
	1.777 ± 0.083	1.19 ± 0.03	1.19 ± 0.02	1.19 ± 0.01
TH-c-4 418.2	0.116 ± 0.010	2.12 ± 0.02	2.14 ± 0.04	2.13 ± 0.03
	0.473 ± 0.032	2.56 ± 0.03	2.55 ± 0.02	2.56 ± 0.02
	0.758 ± 0.019	2.61 ± 0.02	2.69 ± 0.03	2.65 ± 0.01
TH-c-5 435.1	0.104 ± 0.014	2.31 ± 0.02	2.31 ± 0.05	2.31 ± 0.03
	0.678 ± 0.028	3.03 ± 0.00	3.01 ± 0.07	3.02 ± 0.03
	1.042 ± 0.035	3.45 ± 0.03	3.54 ± 0.04	3.50 ± 0.03
TH-c-7 422.9	0.099 ± 0.008	2.09 ± 0.020	2.09 ± 0.04	2.09 ± 0.02
	0.528 ± 0.020	2.56 ± 0.032	2.58 ± 0.02	2.57 ± 0.02
	0.981 ± 0.030	4.37 ± 0.11	4.31 ± 0.11	4.34 ± 0.05
FH-c-3 371.7	0.089 ± 0.006	2.41 ± 0.03	2.46 ± 0.04	2.44 ± 0.03
	0.527 ± 0.024	2.85 ± 0.03	2.89 ±0.02	2.87 ± 0.02
	1.341 ± 0.034	4.60 ± 0.15	4.79 ± 0.09	4.69 ± 0.03
FH-c-4 365.9	0.108 ± 0.013	2.42 ± 0.02	2.52 ± 0.02	2.47 ± 0.02
	1.018 ± 0.031	3.68 ± 0.05	3.84 ± 0.04	3.76 ± 0.01
	1.909 ± 0.507	6.23 ± 0.05	6.23 ± 0.07	6.23 ± 0.02
FH-c-5 353.9	0.117 ± 0.011	2.21 ± 0.02	2.23 ± 0.04	2.23 ± 0.03
	0.596 ± 0.040	2.45 ± 0.03	2.52 ± 0.01	2.49 ± 0.01
	1.147 ± 0.025	3.03 ± 0.04	3.08 ± 0.03	3.06 ± 0.02

**Table 9 materials-16-03284-t009:** Mass water content, w, average mass change over cycles 3 to 5, Δm, and MBV for the four materials under different conditioning.

	Test 1 Dried Then Stabilized at 23 °C; 50%RH				Test 4 Dried Then Stabilized at 23 °C; 33%RH				Test 5 After Test 4, Stabilized at 23 °C; 50%RH				Test 6 After Test 5, Stabilized at 23 °C; 65%RH			
	w_ini_ [%]	w_fin_ [%]	Δm [g]	MBV [g/(m^2^.%RH)]	w_ini_ [%]	w_fin_ [%]	Δm [g]	MBV [g/(m^2^.%RH)]	w_ini_ [%]	w_fin_ [%]	Δm [g]	MBV [g/(m^2^.%RH)]	w_ini_ [%]	w_fin_ [%]	Δm [g]	MBV [g/(m^2^.%RH)]
GY-c-3	0.16	0.17	0.21	0.51 ± 0.01	0.03	0.13	0.20	0.48 ± 0.01	0.17	0.14	0.19	0.46 ± 0.00	0.23	0.14	0.19	0.45 ± 0.00
GY-c-4	0.17	0.18	0.22	0.55 ± 0.01	0.03	0.17	0.23	0.53 ± 0.02	0.18	0.18	0.21	0.49 ± 0.00	0.25	0.16	0.21	0.49 ± 0.01
GY-c-6	0.15	0.16	0.21	0.52 ± 0.01	0.03	0.19	0.21	0.49 ± 0.01	0.17	0.21	0.20	0.48 ± 0.00	0.23	0.19	0.20	0.47 ± 0.00
CC-c-1	0.79	0.75	0.38	1.24 ± 0.02	0.80	0.87	0.36	1.11 ± 0.01	1.00	0.96	0.37	1.13 ± 0.00	1.32	1.16	0.37	1.14 ± 0.02
CC-c-2	0.78	0.75	0.38	1.22 ± 0.02	0.80	0.88	0.35	1.09 ± 0.01	1.01	0.96	0.36	1.11 ± 0.00	1.32	1.16	0.37	1.12 ± 0.02
CC-c-7	0.81	0.77	0.38	1.23 ± 0.02	0.78	0.85	0.36	1.11 ± 0.01	0.98	0.93	0.36	1.12 ± 0.01	1.30	1.14	0.42	1.13 ± 0.01
TH-c-4	3.01	2.95	0.78	2.13 ± 0.03	1.83	2.12	0.73	1.89 ± 0.01	2.79	2.70	0.73	1.91 ± 0.01	4.75	4.40	0.78	2.02 ± 0.01
TH-c-5	3.07	3.02	0.79	2.31 ± 0.03	1.80	2.09	0.75	2.10 ± 0.01	2.75	2.66	0.74	2.08 ± 0.01	4.67	4.32	0.79	2.19 ± 0.01
TH-c-7	3.05	2.99	0.77	2.09 ± 0.02	1.78	2.08	0.74	1.93 ± 0.01	2.74	2.65	0.74	1.91 ± 0.01	4.70	4.34	0.78	2.01 ± 0.01
FH-c-3	1.85	1.76	0.88	2.44 ± 0.03	1.55	1.86	0.81	2.16 ± 0.01	2.37	2.24	0.81	2.15 ± 0.01	3.25	2.77	0.87	2.29 ± 0.02
FH-c-4	1.96	1.86	0.90	2.47 ± 0.02	1.56	1.87	0.84	2.19 ± 0.01	2.38	2.26	0.83	2.16 ± 0.00	3.47	2.98	0.88	2.28 ± 0.02
FH-c-5	2.05	1.96	0.88	2.23 ± 0.03	1.55	1.87	0.82	1.99 ± 0.01	2.38	2.25	0.80	1.96 ± 0.01	3.49	2.99	0.87	2.11 ± 0.01

## Data Availability

For the data supporting, please contact the corresponding author.

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
