# Peer review of "Effect of Air Velocity and Initial Conditioning on the Moisture Buffer Value of Four Different Building Materials"

_materials, 2023, doi:10.3390/ma16083284_

Round 1

Reviewer 1 Report

This paper deals with the hygroscopicity of four building materials, which is interest and is comparable to that of foods and cereal grains. The authors used constant RH air with different velocity to wet the materials (75% RH-8hrs) and then to dry the materials (33% RH-16hrs) at 23°C.

(1)   Which quality index like crack or damage should be used to evaluate for the materials?

(2)   Why not use mathematical models such as Page model to fit for the changes in material mass to time?

Author Response

Dear Reviewer,

Thank you for your comments and suggestions. Please find below our answers to your comments :

Our paper was revised by an English speaking people.

  1. Which quality index like crack or damage should be used to evaluate for the materials?

We assume that this question is related to the drying temperature. The drying temperature follows the recommendation of NF EN 12860 for Gypsum and of RILEM TC 236-BBM for other materials. It is supposed that the materials are not damaged. Thus, we didn’t check the presence of crack or damage.

If this question is related to the mechanical behavior, this subject is not investigated in this paper.

  1. Why not use mathematical models such as Page model to fit for the changes in material mass to time?

The purpose of this paper was to study the effect of air velocity and initial conditioning on the MBV of mineral and bio-based materials from experimental point of view. This is the first step of the whole study. It would be interesting to fit the changes in material mass to time. Our objective in future numerical work is to model the heat and mass transfer by using our own TMC code.

We hope this will come up to your expectations.

Yours faithfully,

Reviewer 2 Report

In this manuscript, the authors measured the moisture buffer value (MBV) according to the NORDTEST protocol and to study the effect of air velocity and of initial conditioning on the MBV results for different materials. Two mineral and two bio-based materials are studied. The results demonstrated that GY is a moderate hygric regulator, CC is good, TH and FH are excellent following the NORDTEST classification. I am pleased to recommend the publication of this work after some minor revisions as indicated below.

1.     There are some spelling/grammar errors in the manuscript, such as 300cm2 should be 300 cm2 in line 97. The paper needs a careful revision before resubmission.

2.     In line 141, the legend of figure 3 was missing. Please add accordingly.

3.     Please describe whether the total porosity of the materials is related to the hygroscopicity properties.

Author Response

Dear Reviewer,

Thank you for your comments and suggestions. Please find below our answers to your comments :

  • There are some spelling/grammar errors in the manuscript, such as 300cm2 should be 300 cm2 in line 97. The paper needs a careful revision before resubmission.

We appreciate your thorough review and apologize for any confusion caused by these mistakes.

The paper was revised by an English speaker to avoid spelling and grammar errors. The space errors between unit and number were corrected.

  • In line 141, the legend of figure 3 was missing. Please add accordingly.

 The legend was added.

  • Please describe whether the total porosity of the materials is related to the hygroscopicity properties.

The hygroscopic properties are mainly related to the open porosity, thus we have added the measurement of this value in section 2.2.2. Comments were added in the Results section to relate open porosity to MBV.

We hope this will come up to your expectations.

Yours faithfully

Reviewer 3 Report

The paper is overall well written but I am struggling to find originality. Similar studies have been done in the past (not long ago) and I suggest to go back to the literature and better look for these references both regarding air velocities and initial conditions on MBV. I find also conclusion quite superficial. To make this paper up to standard I would expect more studies about the impact of air velocities on materials. 

More in details:

introduction should be improved. Please look at other studies on the topic and make a stronger case on the originality and importance of this paper. 

Line 56 is a repetition of what you said earlier so I would suggest to move it when you talk about MBV in general. 

Line 102 please describe your climatic chamber and better explain the test set up. 

Line 120: please provide more info about  how many hours you preconditioned the specimen. How much time did it pass between one test and the other? There is a debate about pre-conditioning so it is important to specify if you weighted the specimens to check if the weight stabilised before testing or you just set a certain time (e.g 24hours)

Author Response

Dear Reviewer,

Thank you for your comments and suggestions. Please find below our answers to your comments :

The English was revised by an English speaking people.

  • Introduction should be improved. Please look at other studies on the topic and make a stronger case on the originality and importance of this paper.

The introduction was improved. We have added 3 references and highlighted the objective and the originality of the paper.

The conclusion was also more detailed.

  • Line 56 is a repetition of what you said earlier so I would suggest to move it when you talk about MBV in general.

The paragraph was moved as suggested.

  • Line 102 please describe your climatic chamber and better explain the test set up.

The end of the section 2.1.1 was rephrased to answer this comment.

  • Please provide more info about how many hours you preconditioned the specimen. How much time did it pass between one test and the other? There is a debate about pre-conditioning so it is important to specify if you weighted the specimens to check if the weight stabilized before testing or you just set a certain time (e.g. 24hours)

As state in the protocol, all the specimens were weighed until reaching the stabilization criteria as mentioned in equation (2). Section 3.1.1 was added in order to detail the stabilization times of four materials at different initial conditions.

We hope this will come up to your expectations.

Yours faithfully,
